# Pitting Influence on Electrical Capacitance in EHL Rolling Contacts

Anatoly Zaiat *,† , Karim Ibrahim † and Eckhard Kirchner

Product Development and Machine Elements, Technical University of Darmstadt, Otto-Berndt-Straße 2, 64287 Darmstadt, Germany; Karim.k.ibrahim@web.de (K.I.); kirchner@pmd.tu-darmstadt.de (E.K.)
* Correspondence: zaiat@pmd.tu-darmstadt.de
† These authors contributed equally to this work.

**Abstract:** This work presents an investigation on the influence of pitting in EHL rolling contact. The pitting geometry comes as an extension on the architecture for multi-physical numerical calculations of rolling element bearing contacts based on OpenFOAM. The model utilized is built according to the state-of-the-art for fluid–solid interaction and electro-quasi-static fields. In this framework, the contact is reduced to be two dimensional in order to reduce the computing costs needed. The changes in the electric properties, which are caused by pittings, are measured using the electric capacitance of the contact and put in perspective with regard to the EHL contacts geometry. The investigation delivers an evaluation on how surface degradation, in both the pitting width $w$ and pitting depth $d$ of the contact, affects the electric properties. It can be seen that the maximum deviations with different reduced radii for the same pitting structures are around 5% and would therefore hardly be distinguishable with corresponding measurements. By comparing the same data for the same ratio of pitting width to reduced radius, on the other hand, differences of up to 50% can be seen.

**Keywords:** EHL (elasto-hydrodynamic lubrication); FSI (fluid–solid interaction); tribology; rolling bearing; condition monitoring; predictive maintenance

## 1. Introduction

The common method used in the industry to estimate the rating life of a rolling bearing follows the prescribed model provided by the international standard ISO 16281 [1]. This method follows a statistical model that is primarily based on the work carried out by Lundberg and Palmgren [2]. The model only considers the rolling contact fatigue as a cause of damage. Other damage mechanisms such as wear and corrosion are neglected, as are multiple factors such as electric current, water in lubricating oil, and hydrogen embrittlement [3]. Hence, due to the statistical nature of the model, rolling bearings are often replaced prematurely, thus causing an increase in downtime and the overall maintenance costs.

The phenomenon of electric current-induced damage affects all power-dense systems. The rise of electric mobility means that this type of damage in the powertrain of electric vehicles has greater significance for the industry. The electric damage in such systems is caused by the voltages induced from the inverters, which are needed to convert direct current into alternating current [4]. Additionally, a voltage difference between the drive-shaft and the motor housing induces damaging mechanisms like discharges through the tribological contacts in the load zone of adjacent bearings [5,6]. The degree of harmfulness of bearing currents can be determined using the apparent bearing current density, which is the electric current passing through the bearing divided by the Hertzian contact area. Such a phenomenon is investigated in the work carried out by Muetze [7], where the author defines threshold values for the current density below which no electric damage occurs ($J < 0.1 \, \text{A/mm}^2$). Therein, the apparent current density $J$ is defined by the ratio of applied

electric current and the Hertzian contact area $A_{hz}$. Therefore, such harmless electric signals can be applied and are considered suitable for sensory utilization [8,9]. Ergo, an approach can be derived to be able to use the change in the electric signal for the purposes of using a rolling bearing as an in situ sensor and thus prognose the remaining operating life of rolling element bearings. The advantage of this method is the direct monitoring of the machine element, the damage location, and the lubricant's condition [10] without further structural components in the chain of action that can influence the measurement.

At the same time, it is known from the research of, for example, de La Guerra Ochoa [11] and Marian [12,13] that regular surface textures in rolling bearings can lead to an increase in the mean lubricant film thickness. The two authors attribute their results to fluidic effects that depend on the geometry of the contact partners and on operating conditions. Similar effects are also known from hydrodynamic thrust bearings [14]. The occurrence of individual pitting structures, which usually has negative effects on the lubricant film thickness, could therefore also have positive effects in individual cases.

### 1.1. Analytical and Numerical Models

Approaches to model the electrical behavior of rolling bearings have existed for several years. One of the first analytical models was described by Prashad in 1988 [15], for example, and later developed further semi-analytically by Puchtler [16] and Schneider [17]. These description models have analytical or empirical conditions in common, such as an ideal surface of the contact partners or a purely radial load on the contact. Major contributions in this field can be attributed to the work carried out by Dowson and Higginson [18] or later Hamrock and Dowson [19]. Still, deviations from these requirements, for example, irregular or damaged surfaces or combined load conditions, cannot be represented with such models. Numerical models of the elastohydrodynamic lubrication (EHL) contact, however, are not bound to such simplifications. Work by Almqvist [20], Neu [21], or Singh [22] has already been able to overcome this limit of ideal contact surfaces. Although the duration of a numerical calculation is several orders of magnitude longer that of analytical and semi-analytical models, the advantage of higher accuracy and more versatile boundary conditions becomes increasingly usable with increasing computing power.

A major advantage of numerical models is the independence of the surface under consideration. While analytical models can hardly or only consider certain regular surface changes, the geometry of the bodies is much more variable in numerical models. Already in ideal equations of state like the numerically solvable Reynolds equation, originally derived for hydrodynamic journal bearings, the description of the fluid film is possible, including in irregular surfaces, see [23]. However, this can only be used to represent changes in the contact partners along the film thickness. Numerical computational fluid dynamics (CFD) models, on the other hand, overcome this disadvantage. This is especially obvious to see in the work carried out by Singh et al. [22], in which the influence of an angular surface crack is investigated. Thus, numerical models can be used to investigate, for example, the influence of pittings and their initiated cracks.

### 1.2. Scope and Assumptions

This paper builds upon the work carried out by Neu et al. [21,24] in the field of EHL and aims to further bridge the gap between the numerical data, which has been known for several decades, and the recent experimental data. The CFD model developed by Neu et al., in the simulation environment OpenFOAM, simplifies the rolling bearing as an analogous model of a parallel plate capacitor, as in previous experiments [15,25]. In this model, the inner raceway and the rolling element represent the parallel plates of the capacitor, while the lubricant is considered to be the dielectric material, as shown in Figure 1 and explained in the simplest form as a plate capacitor in Equation (1).

$$C = \epsilon_0 \epsilon_R \frac{A_{hz}}{h_c} \qquad (1)$$

To describe the electric capacitor, the mean lubricant film thickness $h_c$, calculated numerically or empirically; the Hertzian contact area $A_{\mathrm{hz}}$; and the electric constant $\epsilon_0$ with an approximate value for the relative permittivity $\epsilon_{\mathrm{R}}$ are usually used.

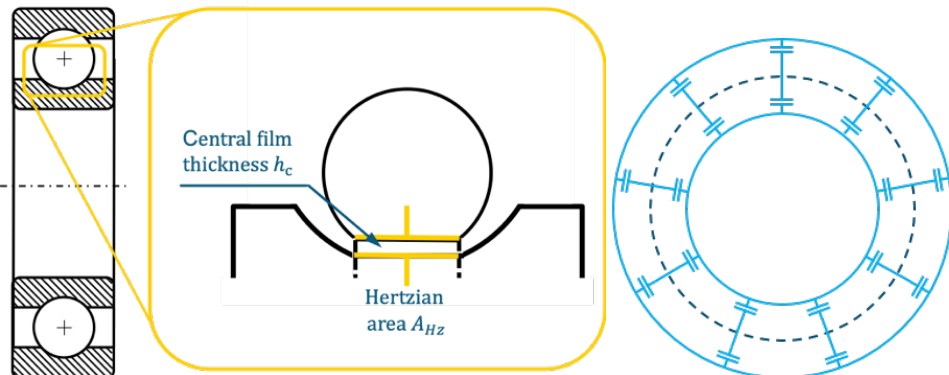

**Figure 1.** Left: Capacitor model of the EHL contact from Martin [9]. Based on the work carried out by Gemeinder [8]. Right: Electrical model representation of the rolling element bearing.

A long-term rolling bearing damage mechanism is the occurrence of pittings. To counteract these, rolling element bearings are either oversized or replaced at an early stage. Hence, it is crucial for the bearing life and for condition monitoring to be able to detect such damage as reliably as possible, improving resource efficiency.

### 1.3. Experimental Findings

Previous experimental work carried out by Martin et al. [9] has already confirmed the suitability of rolling bearing impedance measurement for damage detection. Fatigue tests were carried out, and the impedance of the rolling bearing was recorded and examined at regular intervals, see [9]. It was shown that both the ohmic and capacitive components of the impedance change when pittings are rolled over, see Figure 2, and that the length of the signal can be quantitatively inferred from the length of the damage. It is also shown qualitatively that the pitting depth has an influence on the strength of the signal, although a precise correlation could not be established satisfactorily due to the statistical scatter of the pittings.

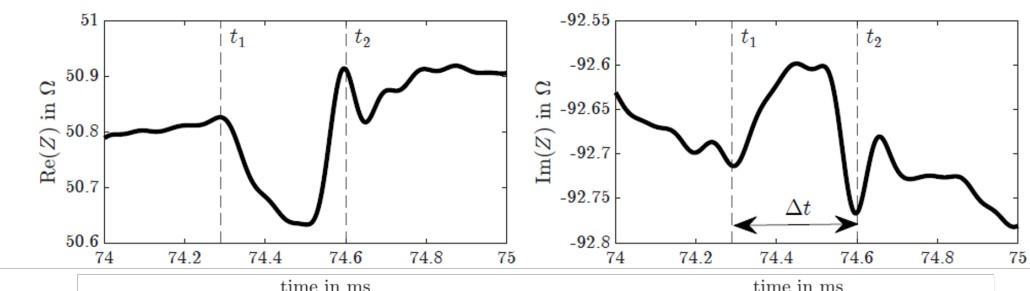

**Figure 2.** Duration of an ohmic and impedance signal deviation indicating a pittings length, following [9].

Furthermore, a change of the electric impedance over lifetime was observed and statistically assessed [26], as shown in Figure 3. Since the research target is to correlate the change in impedance to the remaining useful life of the rolling element bearing, it is essential to understand the correlation between a change of surface properties and the variation of electric properties. As per the state of research of rolling element bearings in electric circuits [27], there is no numerical model being capable of predicting a change in impedance caused by a change of surface properties.

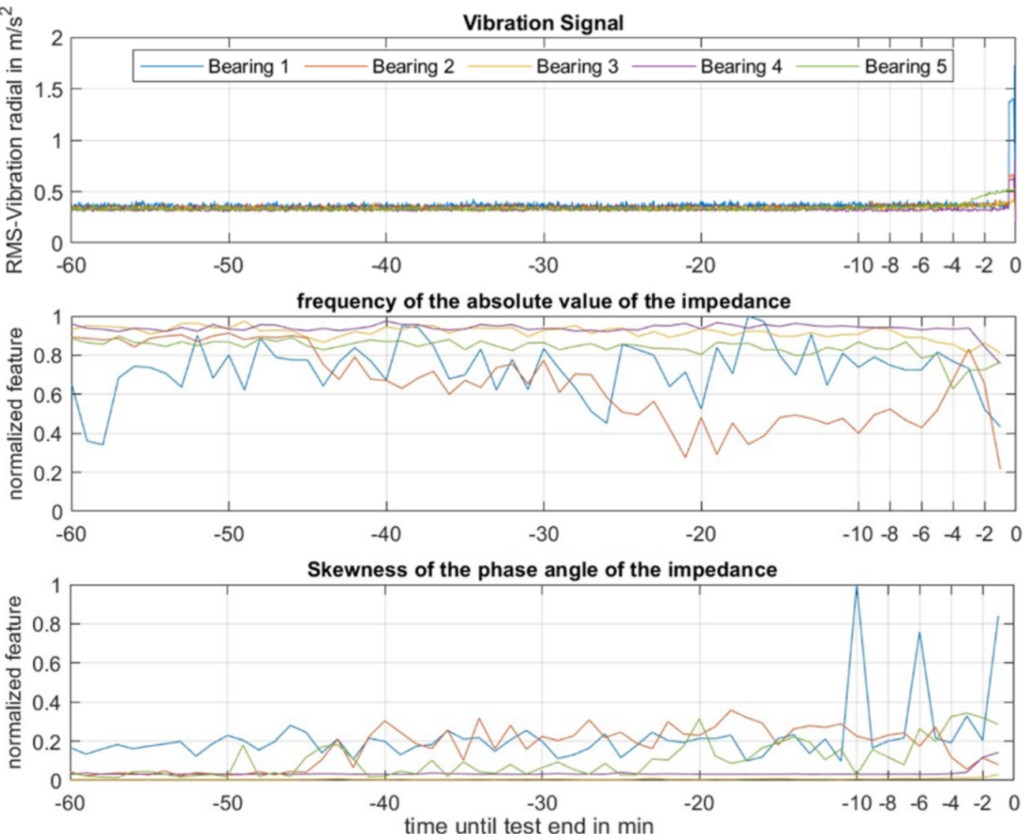

**Figure 3.** Vibration signals and impedance features in the last 60 min of the fatigue test of a rolling element bearing [26].

Therefore, in the framework of this paper, the model developed by Neu et al. [21] is utilized and extended to accommodate for defined surface irregularities in the form of pittings to investigate the experimentally shown impact a pittings depth has on the impedance signal. Furthermore, a pittings geometry is compared to the overall reduced radius of the contact $R_r$. The effect of pittings on the EHL contact is investigated, with a particular focus on the changes observed in the electric properties of the contact. Knowledge of the relationship between changes in electric capacitance and surface geometries, in combination with the rolling bearing impedance measurement, thus leads to more accurate condition monitoring of rolling bearings and forms the core of this paper. A validation with the empirical equation of state of Dowson and Toyoda [28] has already been performed in a previous work by Neu et al. [21].

## 2. Materials and Methods

The calculation of EHL systems such as rolling bearing contacts is generally regarded as a multi-physical phenomenon and is classified as a fluid-structure interaction (FSI) problem. Several source codes have emerged for the numerical calculation of EHL contacts over the past two decades. Starting with the calculations by Almqvist et al. [20] over the numerical extensions made by Hartinger et al. [29] and later by Hajishafiee et al. [30], phenomena such as cavitation and temperature distributions inside the surfaces and rolling elements have been added. Although these approaches exist, the numerical solution of EHL problems is still challenging in terms of calculation time and convergence, especially if additional domains are added. Approaching the described problem, the main FSI coupling is extended with a thermodynamic and electrical domain as shown in Figure 4.

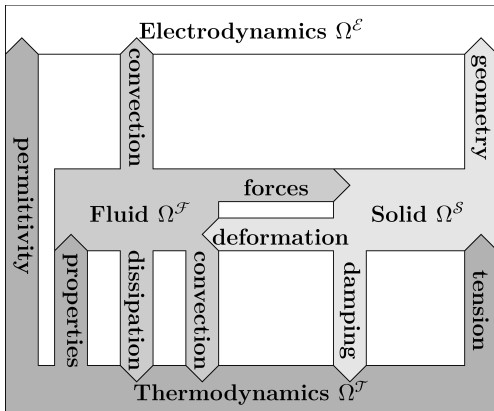

**Figure 4.** Schematic overview of the domains involved [21].

The coupling between the fluid domain $\Omega^F$ and the solid domain $\Omega^S$ is realized through the pressure and drag forces that the fluid imposes on the solid, which in return is deformed, thus changing the geometry of the fluid domain. While high shear rates are present at the inlet of the contact zone, viscous heating contributes to the thermodynamics $\Omega^T$ of the system. In turn, the properties of the fluid, namely, the dynamic viscosity $\mu$ and the density $\rho$ change with temperature $T$ and pressure $p$. The impact extends towards the permittivity $e$ of the fluid.

In this context, surface irregularities of the solid domain produce an effect on the load-bearing capacity of the lubricating film of the fluid domain. The resulting reduced central film thickness $h$ leads to a higher electric capacitance $C$ of the modeled capacitor, resulting in a change in the impedance signal. Note that in this context, the film thickness is calculated numerically and was validated in previous work [21] with the empirical film thickness equation developed by Dowson and Toyoda [28].

The describing differential equations and analytical terms for the material properties are given in the following.

### 2.1. Fluid Domain

The basic differential equations for a compressible calculation of the fluid domain are the mass transport in Equation (2) and the momentum transport in Equation (3).

$$\frac{\partial \rho}{\partial t} + \nabla \cdot (\rho \mathbf{u}) = \mathbf{0} \tag{2}$$

$$\frac{\partial}{\partial t}(\rho \boldsymbol{u}) = -(\nabla \cdot \rho \mathbf{u}\mathbf{u}) - \nabla \rho - \nabla \cdot \boldsymbol{\tau} \tag{3}$$

The rate of change in density $\rho$ is governed by the mass flux $\rho\boldsymbol{u}$. The description for the corresponding compressible momentum transport, Equation (2), is given by [31]. Herein, the temporal rate of change in momentum $\rho\boldsymbol{u}$, on the left-hand side, is attributable to the convective transport of impulse $(\nabla \cdot \rho\mathbf{u}\mathbf{u})$ and the pressure gradient $\nabla\rho$, and the divergence of the shear stress tensor $\nabla \cdot \boldsymbol{\tau}$. Volumetric forces like gravitational acceleration are neglected throughout this work. The stress tensor $\boldsymbol{\tau}$ is specified according to the generalized Newton's law of viscosity assuming isotropy for the fluid [31].

The compressible calculation of the fluid domain requires an equation of state (EOS) to set pressure temperature and fluid density into context. In this case, the Tait EOS in Equation (4) is used.

$$\frac{V}{V_0} = \frac{\rho_0}{\rho} = 1 - \frac{1}{1 + K_0'} ln(1 + \frac{p}{K_0}(1 + K_0')) \tag{4}$$

$$K_0 = K_\infty + \frac{K_0'}{T} \tag{5}$$

The change in specific volume $V$ relative to the volume $V_0$ at ambient conditions is described based on the bulk compressibility modulus $K_0$ at ambient pressure $p_0$ and its rate of change with pressure $K_0'$. For the current work, the linear description by Bair et al. [32] has been chosen to establish a temperature dependence according to Equation (5) because it is considered suitable for all temperature ranges.

Equation (4) is accurate across all pressure regions and is extended to the temperature through the bulk modulus according to Fakhreddine et al. [33]. The Tait EOS is seen to be the most accurate representation for density unless no empirical equation is used, according to Bair [32]. Building on this, the Doolittle equation [34], ref. [35] is used to describe the viscosity over the pressure and temperature range according to Equation (6).

$$\mu = \mu_R exp \left[ BR_0 \left( \frac{\frac{V_\infty}{V_\infty R}}{\frac{V}{V_R} - R_0 \frac{V_\infty}{V_\infty R}} \right) - \frac{1}{1 - R_0} \right] \tag{6}$$

Apparently, the equation does not depend on pressure or temperature explicitly but rather indirectly via the specific volume $V$ and how it changes with temperature and pressure. Therefore, the Doolittle equation is only applicable in conjunction with an EOS and has been designed to be used with the mentioned Tait EOS from Equation (4). There, $R_0$ is the ratio of the free volume $V_{\infty R}$ to the total volume $V_R$ at a reference state $R$, $\mu_r$ the fluids dynamic viscosity at the same reference state, $V_\infty$ the volume occupied by the fluids molecules, and $B$ the Doolittle parameter for the fluid. The dependence of viscosity on temperature and pressure is expressed implicitly by changes in the specific volume $V$.

In contrast to some recent publications in which a cavitation model was implemented, see [22,29], for example, this paper describes the EHL contact as a single-phase model. The divergent outlet region of the rolling bearing contact is therefore limited to the vapour pressure. The single-phase approach reduces the implementation complexity for the electrodynamics domain as no phase change and boundary layer need to be described.

### 2.2. Solid Domain

For the calculation of the solid domain, the elastic deformation of the contact area, commonly referred to as the Hertzian contact area [36], is described via Equation (7).

$$\frac{\partial^2}{\partial t^2} \rho_s \boldsymbol{D} = \nabla \cdot \sigma + \rho_s \boldsymbol{f_b} \tag{7}$$

The displacement vector $\boldsymbol{D}$ is determined by the acting stresses represented by the Cauchy stress tensor $\sigma$ described by Equation (8) and the volume forces $\boldsymbol{f_b}$. The density of the solid is $\rho_S$. The stress tensor is constructed using the strain $\epsilon$ and the trace of the strain $tr(\epsilon)$. In contrast to the fluid domain, which is governed by a Eularian description, the solid domain is described as Lagrangian, leading to the known arbitrary Lagrangian Eularian (ALE) formulation for the coupling problem in FSI. Therefore, the displacement of the volume elements is calculated via the Cauchy stress tensor and probable external volume forces. The Cauchy tensor is defined according to Hooke's law as

$$\sigma = 2\mu_s \boldsymbol{\epsilon} + \lambda_s tr(\boldsymbol{\epsilon})\mathbf{I} \tag{8}$$

with the Lamé constants $\mu_s$ and $\lambda_s$

$$\mu_s = \frac{E}{2(1 - \nu)} \tag{9}$$

$$\lambda_s = \begin{cases} \frac{\nu E}{(1+\nu)(1-\nu)} & \text{for plane stress} \\ \frac{\nu E}{(1+\nu)(1-2\nu)} & \text{for plane strain and 3D cases} \end{cases} \tag{10}$$

To reduce the complexity of the solid calculation approach, the deformation of the solid region is mapped onto the rolling element, utilizing the reduced Youngs modulus $E'$ while including the elasticity modulus of the rings $E_1$ and the rolling elements $E_2$, as well as their Poisson ratios $\nu_1$ and $\nu_2$, respectively.

$$E' = \frac{1}{2}\left(\frac{1-\nu_1^2}{E_1} + \frac{1-\nu_2^2}{E_2}\right)^{-1} \tag{11}$$

Since the problem to be investigated is a contact mechanical problem, the inertia forces are neglected, which reduces the basic differential Equation (7). Inserting Equations (8)–(11) into Equation (7) delivers the final form of the differential Equation (12) to solve for the displacement vector $D$,

$$\frac{\partial^2}{\partial t^2}\rho D = 0 = \nabla \cdot (\mu_s \nabla D + \mu_s (\nabla D)^T + \lambda_s I(\nabla D)) + \rho_s f_b \tag{12}$$

This legitimate simplification is advantageous for the stability of the coupling problem as oscillations between the fluid domain and the solid domain are reduced significantly but can still occur and impact the calculation.

### 2.3. Electrostatic Domain

The descriptive equations for the fluid and solid regions are state of the art for numerical calculations in tribology. A description of the electrodynamic part takes place according to the work of Neu et al. [21]. In general, several equations are needed to fully describe the interacting electric and magnetic fields of a system. Thus, the fully electrodynamic description of the electric field would entail increased computational effort or the consideration of electrodynamic effects within the micro/nanofluidics of the lubrication gap.

To determine whether an electrodynamic or quasi-electrostatic formulation is required, the characteristic time $\tau_E$ of the electrodynamic domain is calculated. $\tau_E$ is the time taken by an electromagnetic wave of velocity $c = 1/\sqrt{\mu_e \epsilon}$ (permeability $\mu_e$ and permittivity $\epsilon$) over the characteristic length of the region under consideration $l$ (lubricant film thickness $h_0$). The time constant, compared to the time it takes for the externally imposed electric or magnetic field to change, determines whether a quasi-static approximation is applicable. For AC fields, this is one quarter of the duration of the period $T_{AC}$. This quasi-static assumption in Equation (13) applies to the electric field in tribological contacts and is stored in Equation (14).

$$\tau_E = \frac{l}{c} = \frac{h_c}{c} \ll \frac{T_{AC}}{4} \tag{13}$$

$$\frac{\partial^2}{\partial x^2}(\epsilon \cdot \mathbf{\Phi}) = -\rho_E \tag{14}$$

Equation (14) describes the potential of the electric field $\mathbf{E} = \nabla\mathbf{\Phi}$, which is impacted by the charge density $\rho_E$ inside of the domain.

## 3. Implementation

The calculation procedure evolves around the FSI and is shown in detail in Figure 5. Contrary to regular FSI calculations where macroscopic elastic deformations of the solid are induced, the present calculations originate from a contact mechanic problem. The aim is to achieve an equilibrium of forces between the fluid film load capability $F_p$ and an externally imposed target force $F_{aim}$. The first one is determined by the pressure and elastic deformation of the contacting area, which in turn depends on the initial lubrication film thickness $h_0$.

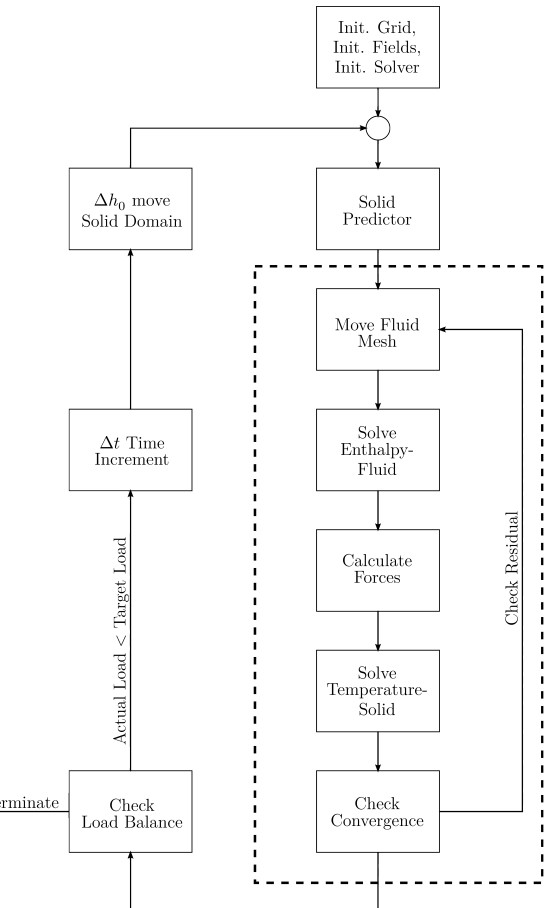

**Figure 5.** Calculation procedure [21].

For the implementation of a load dependent displacement, the approach by Hartinger et al. [29] has been adapted. During the calculation, the initial film thickness is adjusted by the height $\Delta h_0$ according to Equation (15), while the calculation advances in time $\Delta t$.

$$\Delta h_0 = (D_{\max} - D_{\min}) \cdot \frac{F_{\text{aim}} - F_{\text{p}}}{F_{\text{aim}}} \cdot \frac{\Delta t}{t_{\text{cd}}} r_{\text{d}} \qquad (15)$$

In this case, all of the boundaries of the solid domain are moved along the vertical, and the fluid domain is adapted accordingly. Afterwards, the strong coupled iteration between the fluid and solid domain is calculated iteratively until changes in the handover parameters (force $F_p$, deformation $\mathbf{D}$) drop below the convergence criterion.

The change in the initial film thickness $\Delta h_0$ is determined via the deformation difference $D_{\max} - D_{\min}$, the load fraction $(F_{\text{aim}} - F_{\text{p}})/F_{\text{aim}}$, and the time fraction $\Delta t/t_{cd}$, where $t_{\text{cd}}$ is a time constant derived from the speed of sound of the solid domain. The equation is relaxed with the same factor $r_{\text{d}}$ used for the FSI coupling. Even though this equation accounts for the main aspects of time-stepping, deformation, and load, it still needs limitation towards a maximum deformation per time increment to stabilize the iterative procedure. Up until now, the respective value $\Delta h_0 \leq \Delta h_{\max}$ has been found iteratively, and reliable values are provided for the respective calculations.

### 3.1. Numerical Setup

The calculation of multi-physical FSI problems is resource-demanding, even for two-dimensions, such that the three-dimensional representations are typically out of scope, as in this case. Moreover, a three-dimensional calculation would not offer more insights into the topic discussed in this contribution. Therefore, the total calculation domain $\Omega$ is shown in Figure 6 on the right side. It is divided into the solid and fluid subdomain. The fluid region

also contains the thermodynamic domain as no heat transfer into the solid is calculated, as well as the electrostatic domain $\Omega^E$. The coupling of fluid and solid takes place at the interface $\Gamma$, which represents the EHL contact and its transition towards the fluid far field.

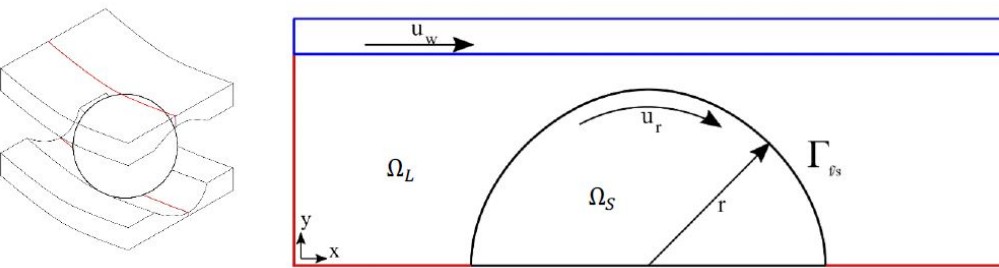

**Figure 6.** 2D simplification of EHL contact [24].

*3.2. Reference and Modified Case*

The process of transferring the roller bearing geometry into the simulation environment of OpenFOAM must be executed in a manner that is optimum for the prevailing flow conditions. However, certain simplifications have to be made in order to ease the complexity and the computational effort. In the framework of academic EHL simulations, such as the work carried out by Almqvist et al. [37], Neu et al. [24], and Hajishafiee et al. [29], a geometrically unified case setup is used. In the model, the geometry of the contact is reduced to a two dimensional numerical grid. Due to the solver being based on the finite volume method, the numerical grid generated is a pseudo-2D grid with one element in the depth direction. Furthermore, a simplification is made to the raceway as it is considered to be flat, and only a partial rolling element with a reduced contact radius $R_r$ is utilized, as depicted in Figure 6 and Equation (16).

$$R_r = \left( \frac{1}{R_{\text{Cylinder}}} + \frac{1}{R_{\text{Raceway}}} \right)^{-1} \tag{16}$$

According to Hertzian theory, the contact between two curved bodies is modeled as an equivalent contact consisting of a body with the reduced radius $R_r$ and a flat counter body. The reduced radius of 10 mm established in the literature [20,22,29,30] would correspond, for example, to the contact of a cylinder of $R_{\text{Cylinder}} = 6$ mm and an inner ring raceway of $R_{\text{Raceway}} = 15$ mm radius. Transferred to real cylindrical bearings, it is noticeable that this ratio is not common. This is particularly evident compared to a real NU1020 cylindrical roller bearing with a 6 mm cylinder roller radius but a real inner race radius of 56.6 mm.

In the context of this paper, the modeled reduced radius of 10 mm is therefore considered as the upper limit at which the model is verified. Furthermore, the numerical investigations are transferred to smaller reduced radii of 7.67 mm and 5.35 mm. This requires higher computing costs or longer computing times but achieves an EHL contact closer to that of more common rolling bearings.

Next, the rolling element is assumed to be rotating in space, while the raceway moves transitionally. The fluid domain is completely filled with lubricant, which is forced into the lubrication gap by the surface velocity of the two bodies.

To further reduce complexity, the rolling element is modeled to be elastic, while the raceway is considered to be rigid. Again, this is done in accordance to the Hertzian stress theory. Additionally, the inlet and outlet were shifted closer to the center of the EHL zone, resulting in the overall contact length being 35% of the reduced radius $R_r$. In this condition, the computational costs remain low, while a numerical starvation effect, as first described by Hamrock and Dowson [38], is ruled out. To rule out phenomena based on the mesh settings, the cell density is based on previous work carried out by Neu et al. [21].

In this framework, the model shown in Figure 7 is used as an undamaged *reference case*. This model is then extended to accommodate a pitting geometry, whose effect on the electric properties of the EHL contact is investigated. In this work, the geometry was

initially defined using a semi-ellipse that is embedded in the fluid domain, where the long semi-axis represents the pitting width while the short semi-axis represents the pitting depth. However, by removing the *arcs* and making the geometry block-shaped, it is observed that the difference in results is minimal, as shown by the pressure distribution shown in Figure 8, while the overall stability of the mesh is increased. The pressure profile over the length of the EHL contact is shown in Figure 8. In the middle of the contact, there is an inserted pitting that disturbs the pressure profile. This shapes the pressure plateau in the center of the EHL contact and prevents the characteristic Petrusevich peak.

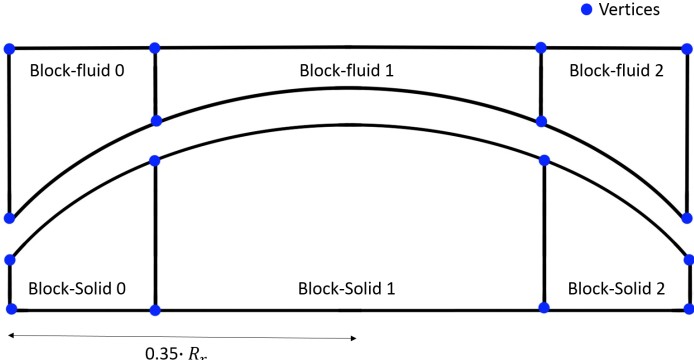

**Figure 7.** Fluid and solid domain of the reference case.

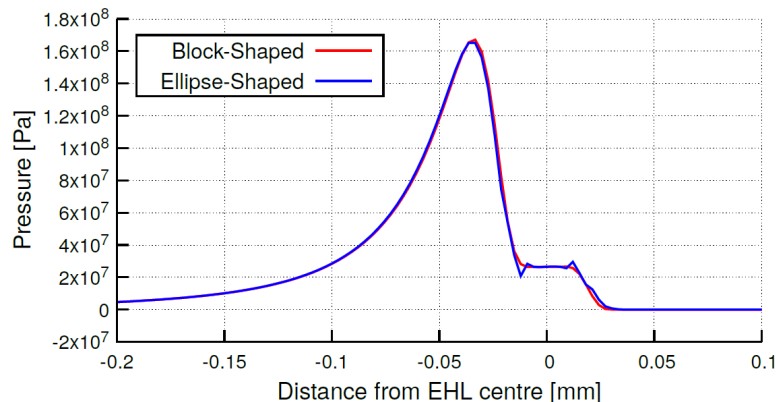

**Figure 8.** Pressure distribution of ellipse-shaped versus brick-shaped pitting geometry in the center of the EHL contact at $\omega$ = 250 rad/s; $T$ = 353 K; $d$ = 9 µm; $w$ = 15 µm; and $R_r$ = 10 mm.

The formed pressure plateau also shows that the pressure in the center of the fitting is approx. 20 MPa and thus above the vapour pressure of the lubricant. In the pittings considered here, there would therefore be no phase transition. Nevertheless, the formation of cavitation phenomena with other surface irregularities is an interesting aspect that should be investigated, especially under the influence of the electrical domain.

Hence, the pitting geometry used throughout the investigation carried out in this paper along with the pitting case are shown in Figure 9. The ratio of the pitting size to the reduced radius $R_r$ shown in Figure 9 is not representative of the actual pittings investigated in this work as the figure only serves as a method to illustrate the geometry. The investigated parameters are given in Table 1, where $R_r$ is the reduced radius, $d_i$ is the pitting depth, and $w_i$ is the pitting width. The ratios of pitting width to the reduced radius $w_i/R_r$ are chosen to represent the spectrum of different pitting geometries. The spectrum ranges from the phenomenon of micropittings, where the intervals are kept at a minimum, to the limit at which the simulations can run stably.

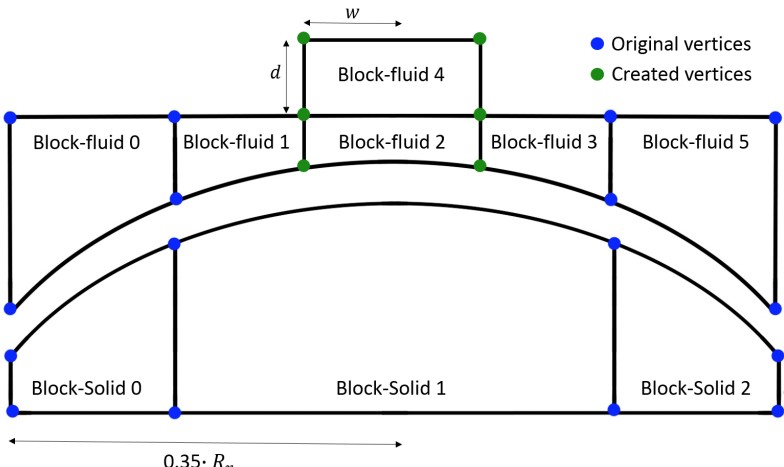

**Figure 9.** Fluid and solid domain of the pitting case.

**Table 1.** Investigated parameters.

| $R_r$ [mm] | $d$ [μm] | $w$ [mm] |
|---|---|---|
| 5.35 | $d_1 = 1$ | $w_1 = 0.0004 \cdot R_r$ |
| 7.67 | $d_5 = 5$ | $w_2 = 0.0008 \cdot R_r$ |
| 10 | $d_9 = 9$ | $w_3 = 0.001 \cdot R_r$ |
| | | $w_4 = 0.0015 \cdot R_r$ |
| | | $w_5 = 0.002 \cdot R_r$ |
| | | $w_6 = 0.003 \cdot R_r$ |
| | | $w_7 = 0.004 \cdot R_r$ |

In the work carried out by Neu [24], it is shown that the mesh of the fluid domain requires a minimum of 8 cells in the vertical contact height in order to obtain valid and stable results. Due to the nature of the investigated parameters, the geometry of the EHL contact is continuously altered between the simulated cases. Hence, in order to maintain a constant aspect ratio between the number of cells in the contact length and the contact height across all the simulations, the cell density in the horizontal contact length of the individual blocks is defined as a linear function of said parameters. In the pitting case, shown in Figure 9, for fluid-block 1, and consequently fluid-block 3, an aspect ratio of 10:1 is implemented in order to maintain the skewness and the stability of the mesh while having reasonable computing time. Fluid-block 2 is considered to be more crucial for the accuracy of the results; hence, a higher cell density is applied since an aspect ratio of 1:1 is used. The cell density of the outer fluid-blocks 0 and 5 remains unchanged from the model developed Neu [24], where it is defined as $15 \cdot R_r$. Furthermore, in order to be able to vary the pitting geometry, the vertices must have the ability to be shifted along the EHL contact.

### 3.3. Boundary Conditions

The boundary conditions implemented in the simulations are based on the experimental work carried out by Martin et al. [9] and the numerical work carried out by Neu et al. [21]. The boundary conditions for density are zero gradient at all boundaries. The pressure gradient is zero at the walls, and pressure is fixed to ambient pressure at the inlet and outlet. There, the velocity gradient is zero. Since the reduced radius $R_r$ is not fixed throughout the simulations, the angular velocity is varied in accordance to the rolling condition, in order to ensure that the same surface velocity is maintained in all the simulated cases.

Another aspect of the boundary conditions is the condition of the remaining surfaces, apart from the pitting itself. As shown in experiments [9], metallic contact occurs at the run-in phase of a rolling element bearings life but is absent once the surface roughness is smoothed out. Additionally, as shown by Srirattayawong [39], the average roughness

of bearings lies one scale beneath the lubricant film thickness calculated in this paper. Therefore, a smooth surface is applicable in the scope of this paper.

　　Furthermore, the parameters of the SAE 20 lubricant [32] are used to model it for the simulations. The boundary conditions used and the SAE 20 modelling parameters are given in Tables 2 and 3, respectively.

**Table 2.** Boundary conditions.

| Parameter | Variable | Value |
|:---:|:---:|:---:|
| Lubricant temperature | $T$ | 333 K[1] |
| Electric potential | $\phi$ | $2.5 \, \text{kg}^1 \text{m}^2 \text{A}^1 \text{s}^{-3}$ |
| Surface velocity | $u_R$ | $2.5 \, \text{m}^1 \text{s}^{-1}$ |
| Youngs modulus | $E_{1,2}$ | $2.1 \times 10^{11} \, \text{kg}^1 \text{m}^{-1} \text{s}^{-2}$ |
| Relative permittivity | $\varepsilon_R$ | 2.10 |
| Oil Density (At 288 K) | $\rho_{oil}$ | $878 \, \text{kg}^1 \text{m}^{-3}$ |

**Table 3.** FVA 3A dynamic viscosity and SAE 20 modelling parameters taken from Bair [32].

| Parameter | Variable | Value |
|:---:|:---:|:---:|
| Doolittle parameter | B | 3.520 |
| $V_{\infty R}/V_R$ | $R_0$ | 0.6980 |
| Thermal expansion of occupied volume | $\varepsilon_t$ | $-1.034 \times 10^{-3} \, \text{K}^{-1}$ |
| Thermal expansion | $a_v$ | $8 \times 10^{-4} \, \text{K}^{-1}$ |
| Dynamic viscosity at reference state $T = 293$ K | $\mu_r$ | $0.1089 \, \text{kg}^1 \text{m}^{-1} \text{s}^{-1}$ |
| FVA 3A dynamic viscosity at reference state $T = 313$ K | $\mu_r$ | $0.0807 \, \text{kg}^1 \text{m}^{-1} \text{s}^{-1}$ |

## 4. Results

　　The magnitude of the effect that the pitting geometry has on the electric properties of the EHL contact is determined using the capacitance of the contact. Based on the simplified analogous electric model of the parallel plate capacitor [8], the capacitance can be calculated using Equation (1). Since the dielectric $\varepsilon$ and the Hertzian area $A_{hz}$ are assumed to be constant throughout the simulations [16], the capacitance $C$ is considered to be solely dependable on the central film thickness $h_c$. Therefore, the capacitance is considered to be anti-proportional to the central film thickness.

　　The effect of the different pitting geometries investigated is shown in Figure 10, where the measured film thicknesses are compared using a fixed reduced radius of $R_r = 7.67$ mm. The results of the performed FSI simulations show that increasing the pitting width causes a decrease in the measured film thickness. By comparing the reference case to the largest simulated pitting case, at a pitting depth of $d_9$, it is calculated that the film thickness drops by about 42%.

　　Furthermore, it is observed that initially for the small to medium sized pitting widths, $w_1$ to $w_5$, the difference between the pitting depths $d_1$ and $d_5$ is minimal. The calculated difference in film thickness ranges between 2.7% and 4.8%. However, the differences between the pitting depths $d_5$ and $d_9$ can be considered negligible for these cases.

　　Upon approaching the upper limit of the pitting geometries that can be stably simulated, $w_6$ and $w_7$, it is observed that the differences between the pitting depths $d_5$ and $d_9$ become more pronounced.

　　This phenomenon is further investigated by comparing the three reduced radii at the highest and lowest ratios $w_1$ and $w_7$, as shown in Figure 11 left and right, respectively. For the smallest pitting width $w_1$, the trend observed prevails in all the reduced radii. The differences between the pitting depths $d_5$ and $d_9$ are negligible. In the biggest pitting ratio $w_7$, the film thickness drops noticeably between the pitting depths $d_5$ and $d_9$ across all

reduced radii. The drop in the film thickness for the reduced radius $R_r = 7.67$ mm and $R_r = 10$ mm lies within similar ranges of 8.2% and 7.5%, respectively. However, the drop in the film thickness for the reduced radius $R_r = 5.35$ mm is only 4%.

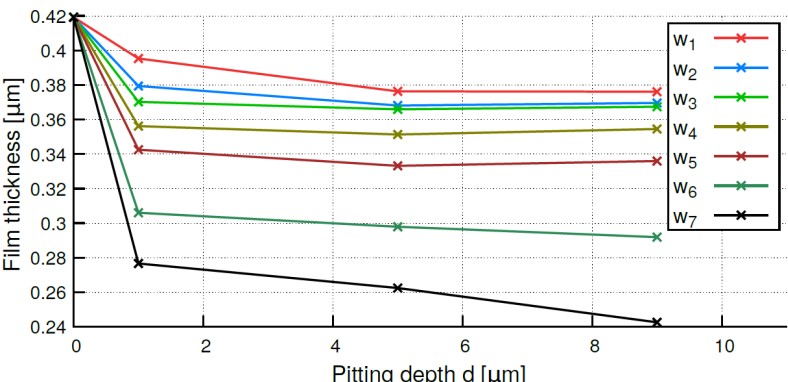

**Figure 10.** Film thickness at different pitting width to reduced radius ratios, at a fixed reduced radius $R_r = 7.67$ mm

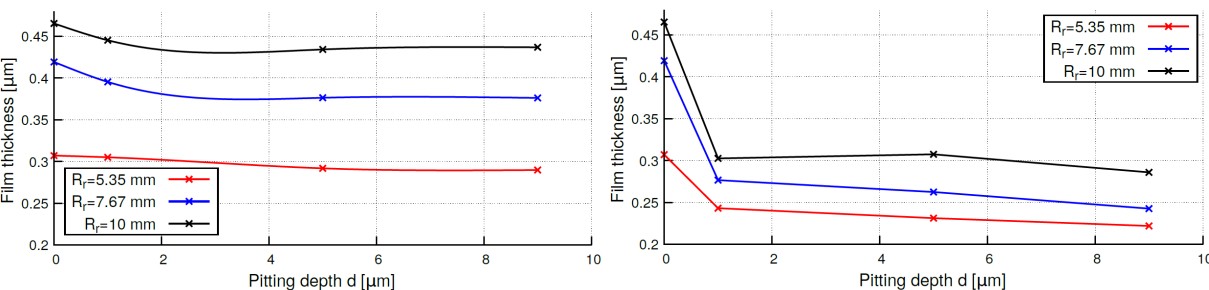

**Figure 11.** Comparing different reduced radii; (**left**) at pitting width to reduced radius ratio $w/R_r = 0.0004$; (**right**) $w/R_r = 0.004$

## 5. Discussion

First, the influence of pittings on the capacitance signal is considered. Figure 12 shows the variation of the capacitance with different, reduced radii $R_r$ and pitting widths $w_i$. The pitting depth $d_5$ is constant with $d = 5$ µm. All of the capacities refer to the lowest capacitance, which occurs at the largest reduced radius $R_r = 10$ mm without a pitting. An expectable progression can be seen here. The smaller the reduced radius, the larger the capacitance. This behavior does not change even if the pittings become wider.

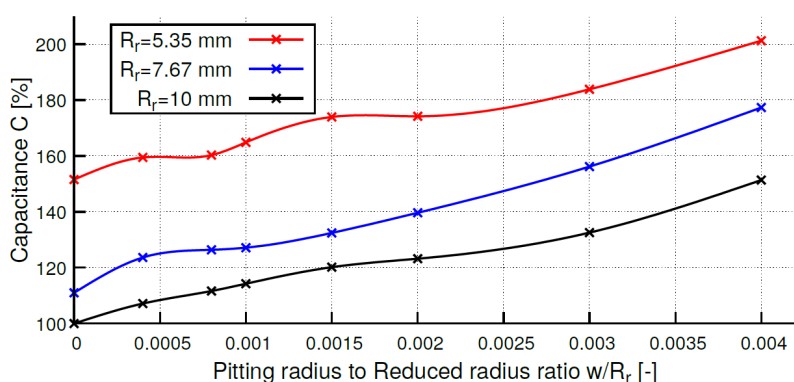

**Figure 12.** Relative capacitance at fixed pitting depth. $d = 5$ µm. All capacitances are relative to the reference case with $R_r = 10$ mm.

Nevertheless, it can already be seen that the gradients show different behavior, which is why all capacitance changes are related to the reference value of the respective reduced radius in the following.

The results show that the electrical properties of the three selected reduced radii respond differently to the simulated surface damage. This is shown both in the absolute view of the pitting geometry and in the ratio of the pitting width to the reduced radius. Since the three reduced radii have different central lubricant film thicknesses due to their geometry, the change in electrical capacitance relative to the reference case of ideal surface is used for this purpose.

First, the influence of absolute pitting widths, at constant pitting depth, on the electrical properties of the simulated EHL contacts is considered. Here, the pitting width, which can be stably modeled, increases with the reduced radius, see Figure 13. Beyond the respective pitting width, the numerical mesh is distorted too much or the contact of both bodies occurs. This can also be observed in the increasing gradient of capacity for wider pitting, indicating a greater reduction in load bearing pressure. Furthermore, it can be seen that for the same pitting width, the change in capacitance is larger for smaller radii than for larger ones. This can also be expected due to the geometric conditions. However, it can be seen that the difference in capacitance change increases from an initial few percent to more than 30% at pitting widths of 30 µm.

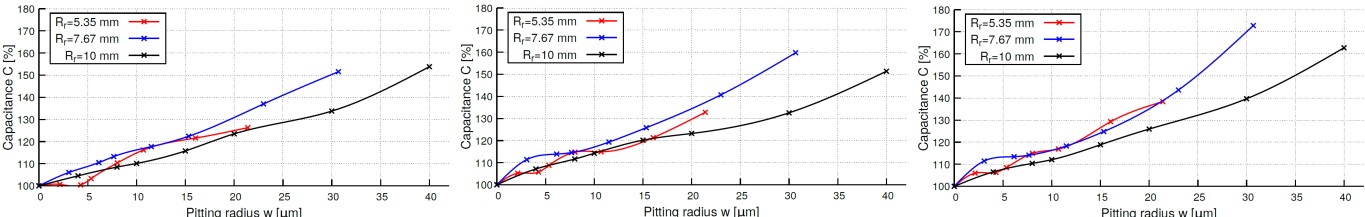

**Figure 13.** Relative capacitance at fixed pitting depth. (**Left**) $d = 1$ µm; (**center**) $d = 5$ µm; and (**right**) $d = 9$ µm. All capacitance changes are relative to the reference case of the individual reduced radius.

Also, a stagnant plateau is evident for all reduced radii, with little change in capacitance for smaller pitting width changes. This is particularly evident in the top and middle graphs in Figure 13, for pitting widths between 5 µm and 15 µm. It can be assumed that the flow of lubricant tends to be turbulent under these boundary conditions and partly compensates for the reduced carrying capacity. The regular machining of the raceway surface to take advantage of this effect had, for example, already been investigated by Marian [12].

It can also be seen that the effects on electrical capacitance are greatest at the medium reduced radius $R_r = 7.67$ mm for the same pitting width at all pitting depths, and smallest at the largest reduced radius $R_r = 10$ mm. In general, once a pitting is present, the change in electrical capacitance is found to be more dependent on pitting width $w_i$ than pitting depth $d_i$. The maximum increase in electrical capacitance in the case of the medium and large reduced radius is 20% at the respective maximum pitting widths of 30 µm and 40 µm. For pitting widths below 20 µm, which can be referred to as micropittings, the difference is less than 5% for all reduced radii considered. It can thus be concluded that pittings with a width of up to 20 µm cause a change in electrical capacitance of about 20–30%, regardless of the size of the rolling element. The pitting depth $d_i$ also plays a rather minor role in this range.

Next, the influence of pitting widths relative to the reduced radius is considered. For this, the same data are used as in Figure 13. Here, a much more uniform progression of the three reduced radii considered can be seen. Despite the compression of the abscissa, it can be seen here that, with the same ratio of pitting width to reduced radius $w_i/R_r$, the capacity change of the medium reduced radius is still just about the largest, see Figure 14.

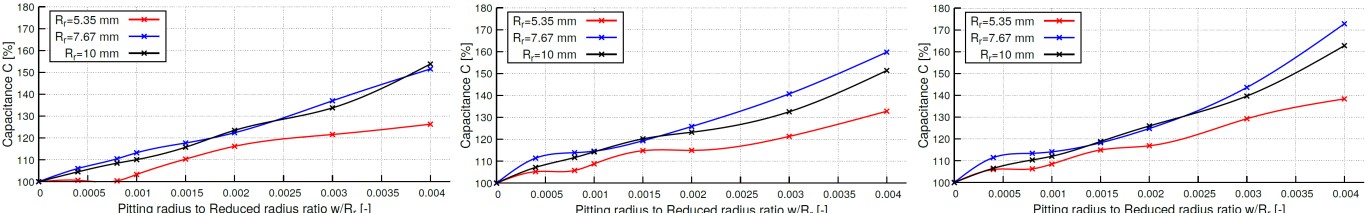

**Figure 14.** Relative capacitance at fixed pitting depth. (**Left**) $d = 1\,\mu m$; (**center**) $d = 5\,\mu m$; and (**right**) $d = 9\,\mu m$. All capacitances changes are relative to the reference case of the individual reduced radius.

In contrast to the absolute plot in Figure 13, the course of the large and mean reduced radius lags are mostly similar and differ by a maximum of 10% for $d = 5\,\mu m$. In contrast to the previous plot, the smallest reduced radius shows the largest deviation from the other two curves, regardless of the pitting depth.

## 6. Conclusions

In conclusion, this paper extends the state of research on the influence of surfaces irregularities on the electrical capacitor model of rolling bearings. The numerical model used herein, benchmarked and explained in more detail in previous works [21,24], is extended to implement different variations of reduced radii and pitting geometries. In a parameter study, the behavior of the electrical capacitor model is investigated in relationship to the modeled reduced radius, the pitting depth, and the pitting width. The operating conditions remain constant.

The changes in electrical capacitance are compared absolutely and relatively to the ratio of pitting width to reduced ratio $w_i/R_r$. The conclusions are as follows:

- The appearance of a pitting-like structure disturbs the lubricant flow, changing the pressure profile over the EHL contact length. Once a pitting is existing, its pitting width $w$ has a more pronounced effect than the pitting depth $d$, as can be seen in Figures 10 and 11.

- As is known from empirical studies [18,19], the geometry of the contact partners also affects the lubricant film thickness. A smaller, reduced radius has a shorter contact length, a higher maximum pressure, and a lower lubricant film thickness. However, if the effects of pitting are compared in terms of the electrical capacitance of the individual EHL contacts, the behavior is approximately the same, see Figure 13. From the knowledge of the geometry of the rolling bearing used, the severity of the surface damage can thus be approximately concluded.

- In the investigated parameter set, it was thus possible to determine that with geometrically identical pittings, the change in capacity lies in a similar range of up to 35% for all reduced radii. Maximum deviations are around 5% and would therefore hardly be distinguishable with corresponding real measurements.

- However, when looking at the same data for the same ratio of pitting width to reduced radius, differences of up to 50% can be seen. This becomes relevant under the aspect that, due to Hertzian theory, the pitting size depends on the contact width. Those deviations would be detectable.

- Still, as shown in Figures 13 and 14, the relative change in the electrical capacitance of the medium reduced radius is similar or up to 10% greater than that of the largest reduced radius. It can be seen that the electrical capacitance of an EHL contact changes similarly for geometrically identical pitting geometries. This behavior is attributed to the interaction between the rolling elements radius and the fluid flow, as investigated in a different manner by Marian [12], but a distinctive relationship is still to be developed.

- It can be stated that regularity can be seen in the behavior of the electric capacitor model, once certain pitting geometries are investigated. As a next step, the model at

hand needs to be enhanced into a three-dimensional model, phase transition needs to be implemented, and the pitting geometries need to be investigated experimentally.

One of the aims of describing the electrical properties of rolling element bearings is to identify damage mechanisms. Work carried out by Almqvist et al. [37] and Singh et al. [22] already presented the influence of surface damages on the EHL state. Marian [12,13], on the other side, was able to show that different changes in topography can influence the EHL state positively. To conclude, this paper connects the disciplines of EHL research and extends them through the electrical capacitance model.

**Author Contributions:** All authors contributed to the study conception and design. Material preparation, data collection and analysis were performed by A.Z. and K.I. The first draft of the manuscript was written by A.Z. and K.I., and all of the authors commented on previous versions of the manuscript. Review and editing was done by E.K. All authors have read and agreed to the published version of the manuscript.

**Funding:** This research received no external funding.

**Institutional Review Board Statement:** Not applicable.

**Informed Consent Statement:** Not applicable.

**Conflicts of Interest:** The authors declare no conflicts of interest.

## Abbreviations

The following abbreviations are used in this manuscript:

| | |
|---|---|
| AC | Alternating current |
| CFD | Computational fluid dynamics |
| EHL | Elastohydrodynamic lubrication |
| EOS | Equation of state |
| FST | Fluid–structure–interaction |

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
