# Peer review of "Pitting Influence on Electrical Capacitance in EHL Rolling Contacts"

_lubricants, doi:10.3390/lubricants11100419_

Round 1

Reviewer 1 Report (Previous Reviewer 1)

1.The originality and innovation of the presented paper are still being concerned, because Section 2 is almost the same with the existing research. The authours should indicate the novelty of the presented paper and its differences from others clearly.

2.Figure 8 shows the EHL pressure distribution, but there is no pressure spike presented yet, which is a typical characteristic of EHL pressure. And why there is a platform around 0 mm? The authors should explain this phenomenon.

3.I'm curious about the treatment of carvitation herein. There is a posibility that carvitation effect would occur in the region of pitting, so how the authours consider this problem. And it seems that there is no description about this boundary condition.

4.The authours indicate "in the context of data-driven damage detection and diagnosis" in Abstract, but it seems that there is no description of "data-driven" in the full paper.

5.The authours should add more data in the Conclusion and Abstract to describe the conclusions more quantitively.

Moderate editing required

Author Response

Reviewer 2 Report (Previous Reviewer 2)

The new submission is basically same with last revision . I have no more questions about the manuscript since the authors have already addressed my main considerations in the last round of peer review.  

Author Response

Reviewer 3 Report (Previous Reviewer 3)

I agree to the publication of this article in its current form, as the paper has undergone improvements and the authors have responded diligently to all my questions and suggestions.

The English is fine.

Author Response

This manuscript is a resubmission of an earlier submission. The following is a list of the peer review reports and author responses from that submission.

Round 1

Reviewer 1 Report

This works presents an investigation on the influence of pitting on the electrical capacitance in EHL. There are some questions about the work shown below.

(1) What is the innovation, or novelty, of the presented paper? The authors mentioned that the work was done based on that by Neu et al. So, what are the different points from that work?

(2) The variables used in the equations are void of description, such as in Eqs. 1, 2, 5, 12, etc. The authors should describe the meaning of each variable employed herein.

(3) The authors adopted the mass and momentum transport equations in depicting the hydrodynamic behavior of the fluid domain. But, in most of the research on EHL, one uses the Reynolds equation instead. So, why the authors adopted these two equations herein?

(4) It seems that the mathematical model presented in Section 2 is rather rough. There are still some important contents not shown, such as the film thickness equation. And the relationship between fluid, solid, and electrostatic domains is vague.

(5) In Figure 6, the pressure peak in the outlet region is not obvious, the authors should explain this. The authors should provide the pitting geometry that is simulated.

(6) It seems that the validation of the model built herein was not given. How can the authors guarantee the model is correct and credible? And the authors only showed the capacitance under different conditions, and the distributions of hydrodynamic pressure and film thickness are neglected. Perhaps, it is a bit hard to analyze the relationship between electric capacitance and EHL.

The details and validations of the model are not provided, I therefore do not recommend it to be published.

Some words and sentences should be refined.

Reviewer 2 Report

An investigation on the effects of pitting on the electrical capacitance of EHL rolling contact was conducted in this study, which can be applied to the damage detection and diagnosis of rolling element bearings. The authors adopted a two-dimensional EHL contact model and the simplified analogous electric model of parallel plate capacitor for studying. The influence of pitting geometric parameters on electric properties was analyzed. Generally, this study fully utilizes the previous results of other researchers and realizes some improvements. However, the authors still need to clarify some considerations as below.

1. Regarding the elastohydrodynamic lubrication state of rolling element bearings, it is easy to produce mechanical contact between the rollers and raceways since it is difficult to realize full-film EHL lubrication. Once mechanical contact occurs, the electrical capacitance probably cannot be monitored. Is it the main limitation of the proposed method and how to relive it?

2. As shown in Fig. 8 and Fig 9, the film thickness is quite small and close to the order of surface roughness. Is it acceptable to ignore the surface roughness in this study?

3. Once the proposed method can be used for damage detection in engineering, how to measure the electrical capacitance of the high-speed rotating components? Is it possible to provide a feasible physical measurement scheme?

4. Regrading Fig. 8, why the appearance of pitting decreases the film thickness. Some researches try to introduce pitting textures in rolling element bearings to improve the lubrication performance (You can search some published papers in Friction Journal). Otherwise, the authors need to clarify the definition of thickness when a pitting exists.

5. The authors used “film height” in this manuscript, it is suggested to use “film thickness” instead.

6. Regarding the caption of Fig. 6, what’s the meaning of “-m”? please remove the index “1” from the units, such as K^1.

7. The authors used simplified analogous electric model of parallel plate capacitor to calculate the electrical capacitance of EHL contact. However, the real state is cambered shape.

please see the fifth comment.

Reviewer 3 Report

The investigation of pitting's influence on electrical capacitance in rolling contacts of rolling bearings within the context of elastohydrodynamic lubrication (EHL) is of significant importance in understanding the complex interplay between surface irregularities and electric properties, providing crucial insights for optimizing bearing performance and reliability.

In this study, the use and extension of the Neu et al. model incorporate surface irregularities (pittings), exploring their impact on the EHL contact and highlighting observed changes in electrical properties.

The work is original and the research is generally well conducted. 

My suggestions to bring more clarity are as follows:

1) The concept of bearing capacitance measurement should be better explained using sketches and figures;

2) The proposed experimental setup for measuring the electrical capacitance of bearings should be validated with the calculation of the remaining useful life (RUL) of the bearings, also validated with experimental results; and

3) A full validation of the obtained results is missing, which is why a major revision of the paper is requested.

The English used in this paper is good. The article is readable and the transmitted idea can be easily understood. 

Round 2

Reviewer 2 Report

The authors have kindly addressed my concerns. 

Reviewer 3 Report

The authors answered all questions and took into account all recommendations and suggestions. In its revised form, the paper is acceptable, so I recommend publishing this work in its current form.

The English is good and clear.